# Sociodemographic Inequalities in Outcomes of a Swedish Nationwide Self-Management Program for Osteoarthritis: Results from 22,741 Patients between Years 2008–2017

**DOI:** 10.3390/jcm9072294

**Published:** 2020-07-19

**Authors:** Erik Unevik, Allan Abbott, Stefan Fors, Ola Rolfson

**Affiliations:** 1Stockholms Sjukhem Foundation, 112 19 Stockholm, Sweden; 2Department of Health, Medicine and Caring Sciences, Division of Prevention, Rehabilitation and Community Medicine, Unit of Physiotherapy, Linköping University, 581 83 Linköping, Sweden; allan.abbott@liu.se; 3Aging Research Center, Karolinska Institutet & Stockholm University, 171 77 Stockholm, Sweden; stefan.fors@ki.se; 4Department of Orthopedics, Institute of Clinical Sciences, Sahlgrenska Academy, University of Gothenburg, 431 80 Mölndal, Sweden; ola.rolfson@vgregion.se

**Keywords:** osteoarthritis, sociodemographic, self-management program, rehabilitation, outcomes

## Abstract

The aim of this study is to investigate if there are educational level and birthplace related differences in joint-related pain, health-related quality of life (HRQoL), willingness to undergo joint surgery, walking difficulties, physical activity level, fear-avoidance behavior before, as well as three and 12 months after participation in a structured self-management program for hip and knee osteoarthritis. Differences in adherence to and use of knowledge from the program were also investigated. An observational national register-based study was performed with a prospective longitudinal design using patient and physiotherapist-reported data on 22,741 complete cases from the National Quality Register for better management of patients with osteoarthritis (BOA) during years 2008–2017. At baseline and after three and 12 months follow-up, higher educational level and being domestic-born was associated with less joint-related pain, better HRQoL, lower willingness to undergo joint surgery, fewer walking difficulties, higher physical activity level, and less fear-avoidance behavior. Foreign born individuals demonstrated higher adherence to exercise and reported better use of the self-management program. The BOA self-management program may require further pedagogical refinement to suit participants of different sociodemographic backgrounds and health literacy. A more patient-centered delivery, sensitive to educational, ethnic, and cultural differences may potentially reduce inequalities in future outcomes.

## 1. Introduction

Today, osteoarthritis (OA) is the fastest growing disability, causing disease globally and affecting over 40 million people in Europe [1,2]. According to guidelines from the Osteoarthritis Research Society International [3], core treatment for OA should consist of multimodal intervention, including education, exercise, and weight control to enable OA self-management, thereby aiming to reduce pain, disability, joint stiffness, and improve health-related quality of life (HRQoL). If non-surgical rehabilitation fails to address this adequately, the patient should be evaluated for total joint replacement (TJR). Despite these guidelines, many patients with OA in the knee or hip have not been offered these alternatives before referral to secondary care for TJR evaluation [4,5].

In light of this background, a Swedish national care program for the better management of patients with osteoarthritis (BOA) was launched in 2006 with the intention that all patients in Sweden with symptomatic hip and knee OA should be offered this treatment [6]. From 2008, health care quality process indicators and patient reported outcome measures for BOA have been routinely collected in the BOA national registry and at the end of 2018, 108,885 patients had been registered in the BOA registry throughout Sweden. Participation in the BOA program has been found to improve HRQoL, self-efficacy, and reduce pain and the intake of medications related to the joint pain among the participants [7,8,9,10].

Disparities in the patient reported frequency of OA-related symptoms and self-efficacy to manage symptoms have previously been reported in the Swedish context based on social determinants, such as educational level and domestic/foreign birthplace [10,11,12]. A recently published study by Kiadaliri and colleagues [11] found that Swedish residents diagnosed with knee OA who had lower educational levels had more frequent pain and worse health-related quality of life (HRQoL) than those with a higher educational level. Higher odds of symptomatic or radiologically confirmed OA as well as willingness to undergo TJR have been confirmed internationally in those with lower education [13,14].

Foreign born individuals make up 19.1% of Sweden’s population [15]. In a cross-sectional study, Soares and Grossi [16] found that the frequency of pain during a week was higher among those born outside Sweden than among those born in Sweden. Similarly, Krupic et al. [12] found that foreign born adults reported worse HRQoL related outcomes attributable to hip OA than those born in Sweden. Furthermore, Olsson and colleagues [10] found that less educated and foreign-born participants in the BOA self-management program reported lower self-efficacy scores at baseline and smaller longitudinal improvements. It can therefore be assumed that, for the BOA self-management program to be able to provide equitable and patient-centered care in the Swedish context, the program needs to be sensitive to educational, ethnic, and cultural differences.

Theoretical frameworks propose that social determinants are likely to play a role in OA outcomes [17]. Furthermore, there have been calls for interventions to improve health care quality for disadvantaged populations with osteoarthritis [18,19]. Yet, it is currently not known whether outcomes from participation in the BOA self-management program are influenced by educational level or by domestic/foreign birthplace (i.e., sociodemographic factors).

### Study Aim

The aim of this study is to investigate if there are educational level and birthplace differences in joint related pain, HRQoL, willingness to undergo joint surgery, walking difficulties, physical activity level, fear-avoidance behavior before, as well as three and 12 months after participation in a structured self-management program for hip and knee osteoarthritis. Adherence to and use of knowledge from the program were also investigated.

## 2. Method

### 2.1. Study Design

This is an observational national register-based study with prospective longitudinal design for routine collected outcomes of the BOA Swedish national care program. The BOA self-management program has been described elsewhere [6] and is outlined in Figure 1. The program is comprised of two components, namely a theory part consisting of a minimum of two theory sessions, and a voluntary exercise part. The first lesson contains information regarding the pathophysiology of the disease and available treatments. The second lesson targets why exercise is important for the treatment of OA, coping strategies, and how to incorporate exercise and self-management strategies relating to symptoms and pain into daily life. The third (optional) session is held by a patient with OA, who has been trained as an OA communicator. In the second component of the BOA self-management program, the patient is offered an individual session with a physiotherapist to customize an exercise program. The patient thereafter has the possibility to perform the exercise program in group sessions supervised by a physiotherapist two times per week for 6–8 weeks or to exercise in another location such as in their home.

### 2.2. Sample Selection and Size

All patients from the BOA register, with a first registration (baseline) between 01,012,008–31,122,017 (*n* = 94,798) were eligible. The BOA register prospectively includes patients who have sought treatment for hip and/or knee pain in all the 21 regions of Sweden, and after a confirmed OA diagnosis (clinical and/or radiographic) have been referred to the BOA self-management program. The following exclusion criteria for participation in the BOA self-management program and registration in the BOA register are applied: suspicion of or confirmed tumor, rheumatoid arthritis, sequel hip fracture, chronic pain or fibromyalgia, TJR within the past 12 months or other knee or hip joint surgery within the past 3 months and inability to read or understand the Swedish language.

A flow-chart for defining the selection and size of the analytic sample for this study is illustrated in Figure 2. Patients that entered the register before the end of September of 2016 were excluded because they would not have had the possibility to answer the one-year follow-up (*n* = 24,375). Those who had undergone TJR or decided to drop out for any other reason for surgery were also excluded (*n* = 15,205). In a final step, participants with missing data from any of the covariates, dependent and independent variables in the present study were excluded (*n* = 32,477). This was done to facilitate a complete case analysis [20] as the size of the remaining sample was more than adequate to satisfy data assumptions of statistical tests. Furthermore, a sensitivity analysis showed that the distribution of age, sex, body mass index (BMI), most affected joint, educational level, and domestic/foreign birthplace were similar for the analytical sample and the excluded sample.

### 2.3. Measures

Patient-reported outcome measures are collected and entered in the BOA registry when the patient visits the physiotherapist at baseline and at 3-months follow-up. Patients also receive a final questionnaire by mail or email after 12 months which is entered into the BOA registry. Health care quality process indicators are reported by the responsible physiotherapist or occupational therapist at baseline and after 3 months. The following variables were extracted from the BOA registry for the current study and are outlined in Table A1 (Appendix A).

Independent Variables
-Which is the highest level of education that you have obtained (compulsory, upper secondary, university)?-Were you born in Sweden (yes/no)?CovariatesAll analyses were adjusted for age, sex, and levels of BMI. Age and sex are self-administered variables by the BOA-register, retrieved by the patient’s personal identity number in the baseline questionnaire. BMI (kg/m^2^) was calculated based on patient’s self-assessed weight (kilograms) and length (centimetres).Dependent Variables
-Mean pain intensity from the patient’s most troublesome joint in the past week was self-reported on a numeric rating scale (NRS), ranging from 0–10 (0 = no pain, 10 = maximum pain) [21].-The EQ-5D-3L is a standardised self-reported generic measurement of health. It covers five aspects of health: mobility, self-care, usual activities, pain/discomfort, and anxiety/depression. Each dimension has 3 levels: no problems, some problems, extreme problems, resulting in 243 possible health states. In this study we used the United Kingdom value set, diverging from −0.594 to 1, where a value of 1 means that a person has no problems with any of the five health domains, and value of 0 means that the person has extreme problems with all five health domains [22].-Do you suffer so much impairment from any joint that you are willing to undergo surgery (yes/no)?-Does your pain cause you difficulties with walking (yes/no)?-Being physical active more than 150 min per week (yes/no)? This was originally an ordinal variable with seven answer options (spanning from 0 min to more than 300 min). Categories 150–300 min and more than 300 min were recoded into “yes”, and all other categories into “no”. This because at least 150 min of physical activity weekly is recommended to prevent all-cause mortality and chronic disease [23].-Fear-avoidance behaviour: Are you afraid that exercise or physical activity will be harmful to your joints (yes/no)?-Participation in theory lessons was assessed by a single question assessed by the physical or occupational therapist, namely “has the patient participated in the theory sessions” (yes/no)?-Number of supervised exercise sessions the patient participated in, assessed and reported by the physical or occupational therapist. This variable had five answer categories spanning from no session to more than 12. But to make the variable easier to overview, responses were categorized into: no exercise session, 1–9 sessions, and 10 or more.-How often do you use knowledge acquired from the self-management program? Originally the variable had six categories but for the purpose of interpretation the variable was recoded into two categories. One category included responses like “every day or several times daily” while the other category included responses like “every week, month, never, or don’t know”.


### 2.4. Statistical Analyses

As a first step in the analyses, we explored the distribution of the sample across the sociodemographic characteristics and outcomes. The results are presented in terms of means and standard deviations for the continuous indicators and numbers and percentages for the categorical indicators.

In the next step, we analyzed the associations between the exposure to sociodemographic factors (educational level or domestic/foreign country of birth) and the BOA self-management program outcomes using multivariate analyses.

The continuous outcomes (EQ-5D-3L and NRS pain) were analyzed by analysis of covariance (ANCOVA) with Bonferroni corrections for multiple comparisons. The results are presented in terms of adjusted means with 95% confidence intervals (CI). Educational level or domestic/foreign country of birth were analyzed as in separate ANCOVA models, where they were used alternatively as a fixed factor or covariate.

The dichotomous outcomes were analyzed using binary logistic regressions. The reported degree of participation in exercise session (3 categories) was analyzed using multinomial logistic regression in order to account for nonlinearities in the associations. The results from these analyses are presented in terms of adjusted odds ratios (OR) with 95% CI.

In addition to the main exposures, education and birthplace, all statistical analyses were also adjusted for age, BMI and the baseline score of the outcome. All statistical analyses were conducted using SPSS for Windows, version 25 on a secure online data access (SODA) service provided by the Centre of Registers, Västra Götaland, Sweden where the BOA national registry is housed.

### 2.5. Ethics

The study was approved by the Regional Ethical Review Board Gothenburg (entry number: 986-17). All participants in the register are informed by the responsible physiotherapist that their participation will be registered in BOA and may be used in research.

## 3. Results

Descriptive sociodemographic characteristics for the analytical sample are described in Table 1 and are in line with distributions reported for the entire BOA database [7]. Descriptive results according to educational level and domestic/foreign country of birth for outcome variables related to pain, HRQoL, willingness to undergo joint surgery, and mobility are reported in Table 2. Furthermore, descriptive results according to educational level and domestic/foreign country of birth for outcomes related to health-related behaviours and adherence to the self-management program are described in Table 3. Multivariate analyses adjusted for age, sex, BMI, and baseline scores are described in Table 4 for outcomes related to pain, HRQoL, willingness to undergo joint surgery, and mobility, and in Table 5 for outcomes related to health-related behaviours and adherence to the self-management program.

Longitudinal outcomes comparing educational levels showed statistically significantly higher levels of HRQoL and physical activity at baseline, three and 12 months, regular use of learned content from the self-management program after three and 12 months and higher levels of participation in 1–9 exercise sessions in participants with higher level of education. Furthermore, statistically significant less NRS pain and willingness for joint surgery at baseline and three and 12 months, as well as less walking difficulties and less fear-avoidance behavior after three and 12 months was observed among participants with a higher level of education.

A comparison of country of origin showed statistically significant higher levels of HRQoL and physical activity at baseline and three and 12 months among domestic born participants. Significantly less NRS pain, walking difficulties and fear-avoidance behavior were reported at baseline, three and 12 months in the domestic born groups. Furthermore, significantly less willingness for joint surgery at baseline, less regular use of learned content from the self-management program after three and 12 months, as well as lower levels of participation in exercise sessions was reported in the domestic-born group.

## 4. Discussion

In line with the study’s aim, investigation of registry data from before the BOA self-management program start, after three months follow-up, and after 12 months follow-up could confirm statistically significant differences in outcomes according to education and domestic/foreign country of birth, adjusted for known confounders. Self-reported pain, HRQoL, levels of physical activity, and walking difficulty showed a persistent pattern where participants with higher education and domestic-born participants reported better outcomes from the BOA self-management program over time.

Our findings that participants with lower educational attainment reported more pain and lower HRQoL at baseline and maintained a higher willingness to undergo joint surgery despite participation in the BOA self-management program is in line with earlier research [11,14,24].

Foreign born participants reported a greater willingness to undergo joint surgery at baseline than domestic born participants, but this difference had vanished at the three and 12-month follow-ups. One possible reason for this could be that foreign-born participants in the BOA cohort on average reported higher levels of OA-related pain. This finding is similar to previous literature from Swedish cohorts for TJR [12], self-efficacy of symptoms [10], and musculoskeletal pain [25]. A possible contributing factor to the higher OA-related pain levels in Swedish immigrants could be the higher prevalence of work in blue-collar professions compared to the domestic-born population [26].

Foreign-born participants participated in more exercises sessions and were more likely to utilize what they have learned from the BOA self-management program when compared to domestic-born participants. This may appear contradictory since, in the current study and similar to previous literature [10,12], foreign-born participants tend to report lower levels of HRQoL and physical activity as well as higher levels of NRS pain, walking difficulties and fear-avoidance behavior at baseline and three and 12 months than domestic-born patients, which are known risk factors to low adherence for rehabilitation [26]. On the other hand, previous research has suggested that higher levels of pain and mobility impairments with associated lower physical activity levels could potentially act as a motivator for exercise to alleviate OA-related impairments [27]. However, the BOA self-management program may also provide foreign-born participants with new knowledge and information related to the OA and its management, where domestic-born participants may have prior knowledge.

Both participants with lower levels of education and those foreign-born reported increasingly higher odds for fear-avoidance behavior at all follow-up’s even though all participants receive the message that physical activity and exercise are safe for the joints. Since earlier research [10] has shown that participants with a lower educational level and foreign-born participants in the BOA self-management program report lower levels of self-efficacy at baseline, and as these groups report more OA symptoms at baseline in this study, it may appear counterintuitive to them that exercise is safe for the joints, even after receiving self-management advice. In addition, poor self-rated health and lower educational level have been associated with lower health literacy among patients with OA [28]. This could potentially explain the observed differences in the effectiveness of reassurance messages in the BOA self-management program. Future qualitative studies should explore why attitudes of fear-avoidance vary across sociodemographic groups.

The BOA self-management program is one of Sweden’s biggest public health interventions, with the potential to diminish sociodemographic inequalities in health. However, sociodemographic inequalities in OA-related health exists after participation in the BOA self-management program. This suggests that the educational intervention within the BOA self-management program may require further pedagogical refinement to suit participants of different sociodemographic background and health literacy, as a more personalized delivery of OA self-management care has been recommended in the current literature [28,29].

To our knowledge, this study provides the largest study sample to date for evaluating non-surgical measures to affect OA health in relation to participants sociodemographic background. This is also the first large cohort study to show that foreign/domestic birthplace is associated with varying adherence to an OA management program. In addition, only two prior studies have assessed if educational level is associated with adherence to OA-related rehabilitation [27,30]. However, this study has several limitations that one must take into consideration when interpreting the results. Limitations include the lack of a control group and the exclusion of 50.2% of the original cohort due to missing longitudinal data as we chose a complete case analysis method. A sensitivity analysis showed that the analytical sample had 4% more participants with university education and 2% fewer participants with compulsory education compared to excluded cases (Table A2, Appendix A). Yet, it is possible that an analytical approach with imputation for missing values may have yielded different results.

Self-reported variables over multiple measurements imply an inherent possibility of response-shift and social desirability bias that one must also take into consideration. Another possible limitation was that knee/hip OA were not analyzed separately, as research has shown that patients with hip OA are less likely to respond to nonsurgical treatment [9]. It is not mandatory to speak and understand Swedish in BOA, as it is possible to undertake the BOA self-management program with a translator. However, this could be considered a potential barrier for foreign born participants and for their response to questionnaire data collection.

## 5. Conclusions

The BOA self-management program may require further pedagogical refinement to suit participants of different sociodemographic backgrounds and health literacy. A more patient-centered delivery sensitive to educational, ethnic, and cultural differences may potentially reduce inequalities in future outcomes.

## Figures and Tables

**Figure 1 jcm-09-02294-f001:**
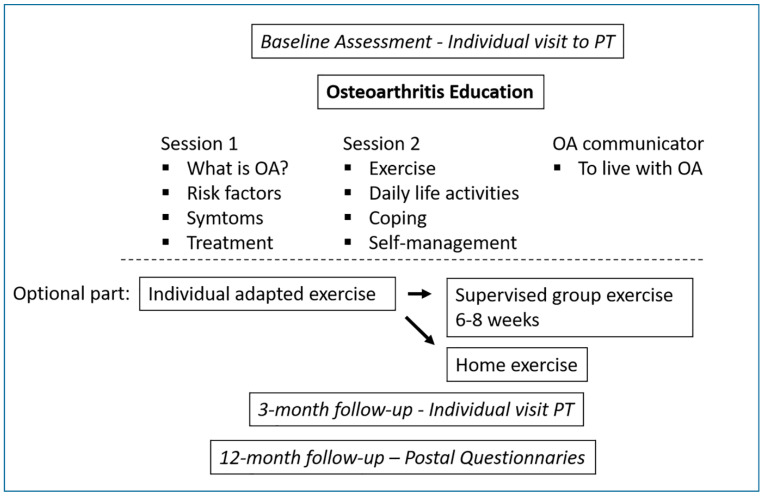
Overview over the education and exercise parts of the BOA self-management program. PT = Physiotherapist. Retrieved with copyright permission from: Jönsson T., Eek F., Dell’Isola A., et al. The Better Management of Patients with Osteoarthritis Program: Outcomes after evidence-based education and exercise delivered nationwide in Sweden. PLoS ONE 2019; 14 (9): e0222657. URL: https://journals.plos.org/plosone/article/figure?id=10.1371/journal.pone.0222657.g001.

**Figure 2 jcm-09-02294-f002:**
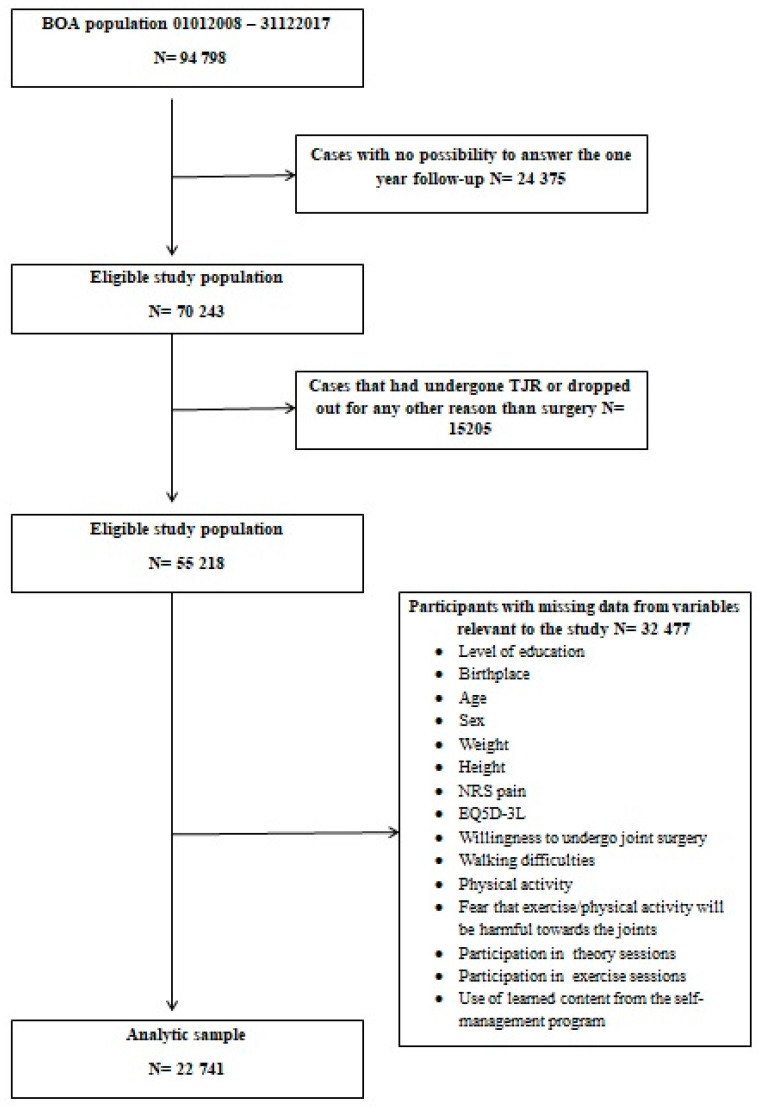
Inclusion and exclusion processes for defining the analytical sample of the study.

**Table 1 jcm-09-02294-t001:** Sociodemographic distribution over featured variables.

	Total (*n* = 22,741)	% (N)	Mean	min, max	SD
Sex	Men	29.3 (6664)			
	Women	70.7 (16,077)			
Educational level	Compulsory	32.2 (7328)			
	Upper Secondary	37.2 (8467)			
	University	30.5 (6946)			
Country of birth	Foreign	7.7 (1744)			
	Domestic	92.3 (20,997)			
Most painful joint	Knee	68.8 (15,677)			
at baseline	Hip	28.5 (6477)			
	Hand	2.7 (607)			
	Age		66.3	27, 95	9
	BMI		27.8	13, 71	4.7

Values are presented in % (*n*), mean & range in numbers, standard deviation (SD).

**Table 2 jcm-09-02294-t002:** Descriptive results for outcome variables related to Pain, HRQoL, willingness to undergo joint surgery and mobility (*n* = 22,741).

NRS Pain (0–10 Scale)								
Level of Education or Country of Birth	Baseline		3 Months		Difference at 3 Months vs. Baseline	12 Months		Difference at 12 Months vs. Baseline
	Mean	SD	Mean	SD		Mean	SD	
Compulsory	5.2	1.8	4.1	2.0	−1.1	4.6	2.2	−0.6
Upper Secondary	5.2	1.9	4.0	2.0	−1.2	4.4	2.3	−0.8
University	4.8	1.9	3.6	2.0	−1.2	4.0	2.3	−0.8
Foreign born	5.5	1.9	4.2	2.0	−1.2	4.8	2.3	−0.7
Domestic born	5.1	1.9	3.9	2.2	−1.2	4.3	2.3	−0.8
Total group	5.1	1.9	3.9	2.0	−1.2	4.3	2.3	−0.8
**EQ-5D-3L**								
**Level of education or country of birth**	**Baseline**		**3 months**		**Difference at 3 months vs. baseline**	**12 months**		**Difference at 12 months vs. baseline**
	**Mean**	**SD**	**Mean**	**SD**		**Mean**	**SD**	
Compulsory	0.64	0.20	0.70	0.18	+0.06	0.66	0.20	+0.02
Upper Secondary	0.64	0.21	0.70	0.18	+0.06	0.67	0.20	+0.03
University	0.67	0.19	0.72	0.16	+0.05	0.70	0.19	+0.03
Foreign born	0.58	0.24	0.67	0.21	+0.09	0.63	0.22	+0.05
Domestic born	0.65	0.20	0.71	0.17	+0.06	0.68	0.19	+0.03
Total group	0.65	0.20	0.71	0.19	+0.06	0.68	0.26	+0.03
**Do you suffer that much impairment from any joint that you are willing to undergo surgery? Yes/No**								
**Level of education or country of birth**	**Baseline** **Yes**		**3 months** **Yes**		**Difference at 3 months vs. baseline**	**12 months** **Yes**		**difference at 12 months vs. baseline**
Compulsory	22.7 (1662)		16.0 (1172)		−6.7	24.6 (1801)		+1.9
Upper Secondary	19.2 (1694)		13.7 (1153)		−5.5	22.0 (1860)		+2.8
University	14.7 (1019)		9.7 (674)		−5	17.4 (1210)		+2.7
Foreign born	23.3 (406)		15.6 (271)		−7.7	24.1 (420)		+0.8
Domestic born	18.9 (3969)		13.0 (2728)		−5.9	21.2 (4451)		+2.3
Total group	19.2 (4375)		13.2 (2999)		−6	21.4 (4871)		+2.2
**Does your pain cause you difficulties with walking? Yes/No**								
**Level of education or country of birth**	**Baseline** **Yes**		**3 months** **Yes**		**Difference at 3 months vs. baseline**	**12 months** **Yes**		**difference at 12 months vs. baseline**
Compulsory	79.5 (5824)		60.9 (4460)		−18.6	63.4 (4644)		−16.1
Upper Secondary	78.5 (6650)		57.4 (4864)		−21.1	59.5 (5042)		−19
University	76.1 (5287)		54.9 (3810)		−21.2	56.1 (3896)		−20
Foreign born	80.6 (1406)		61.7 (1076)		−18.9	63.8 (1113)		−16.8
Domestic born	77.9 (16,355)		57.4 (12,058)		−20.5	59.4 (12,469)		−18.5
Total group	78.1 (17,761)		57.8 (12,134)		−20.3	59.7 (13,582)		−18.4

NRS Pain and EQ-5D-3L are presented in numbers, all other values are presented in % (numbers inside brackets).

**Table 3 jcm-09-02294-t003:** Descriptive results for outcome variables related to health-related behaviours and adherence to the self-management program (*n* = 22,741).

Physical Active more than 150 min Per Week Yes/No					
Level of Education or Country of Birth	Baseline Yes	3 Months Yes	Difference at 3 Months vs. Baseline	12 Months Yes	Difference at 12 Months vs. Baseline
Compulsory	40.8 (2987)	42.2 (3092)	+1.4	37.6 (2753)	−3.2
Upper Secondary	44.9 (3803)	47.7 (4042)	+2.8	44.5 (3764)	−0.4
University	49.8 (3456)	52.6 (3652)	+2.8	50.2 (3487)	+0.3
Foreign born	37.7 (657)	42.4 (740)	+4.7	38.7 (675)	+1
Domestic born	45.7 (9589)	47.8 (10,046)	+2.1	44.4 (9329)	−1.3
Total group	45.1 (10,246)	47.4 (10,786)	+2.3	44.0 (10,004)	−1.1
**Fear avoidance: Are you afraid that exercise or physical activity will be harmful to your joints? Yes/No**					
**Level of Education or Country of Birth**	**Baseline** **Yes**	**3 months** **Yes**	**Difference at 3 months vs. baseline**	**12 months** **Yes**	**Difference at 12 months vs. baseline**
Compulsory	14.3 (1048)	5.5 (402)	−8.8	9.4 (688)	−4.9
Upper Secondary	16.4 (1385)	5.2 (441)	−11.2	8.7 (739)	−7.7
University	15.0 (1043)	4.2 (292)	−10.8	5.7 (393)	−9.3
Foreign born	23.3 (407)	10.8 (189)	−12.5	17.0 (297)	−6.3
Domestic born	14.6 (3069)	4.5 (946)	−10.1	7.3 (1523)	−7.3
Total group	15.3 (3476)	5.0 (1135)	−10.3	8.0 (1820)	−7.3
**Number of supervised exercise sessions the patient participated in**					
**Level of education or country of birth**	**None**		**1–9 Times**		**10 Times or more**
Compulsory	40.6 (2972)		26.9 (1974)		32.5 (2382)
Upper Secondary	42.8 (3625)		28.2 (2391)		28.9 (2451)
University	40.3 (2800)		30.6 (2125)		29.1 (2021)
Foreign born	36.1 (629)		31.8 (554)		32.2 (561)
Domestic born	41.8 (8768)		28.3 (5936)		30.0 (6293)
Total group	41.3 (9397)		28.5 (6490)		30.1 (6854)
**How often do you use acquired knowledge from the self-management program? Every day or several times daily (compared to those who answered: Every week, month, never, or don’t know)**					
**Level of education or country of birth**	**3 months** **Yes**		**12 months** **Yes**		**Difference at 12 months vs. baseline**
Compulsory	60.7 (4450)		36.8 (2700)		−23.9
Upper Secondary	61.3 (5188)		37.4 (3170)		−23.9
University	63.0 (4375)		40.7 (2824)		−22.3
Foreign born	65.1 (1135)		43.2 (753)		−21.9
Domestic born	61.3 (12878)		37.8 (7941)		−23.5
Total group	61.6 (14013)		38.2 (8694)		−23.4

Values are presented in % (numbers inside brackets).

**Table 4 jcm-09-02294-t004:** Adjusted outcomes for variables related to pain, HRQoL, willingness to undergo joint surgery and mobility (*n* = 22,741).

NRS Pain (0–10 Scale) Mean Values						
Education or Country of Birth	Baseline		3 Months		12 Months	
	**Mean**	**CI**	**Mean**	**CI**	**Mean**	**CI**
Compulsory	**5.2**	**5.2–5.3**	**4.0**	**4.0–4.1**	**4.5**	**4.4–4.5**
Upper Secondary	**5.2**	**5.1–5.2**	**4.0**	**3.9–4.0**	**4.3**	**4.3–4.4**
University	**4.8**	**4.8–4.9**	**3.8**	**3.8–3.8**	**4.1**	**4.1–4.2**
Foreign	**5.5**	**5.4–5.5**	**4.0**	**3.9–4.1**	**4.5**	**4.4–4.6**
Domestic	**5.1**	**5.0–5.1**	**3.9**	**3.9–4.0**	**4.3**	**4.3–4.3**
**EQ-5D-3L Mean values**						
**Education or country of birth**	**Baseline**		**3 months**		**12 months**	
	**Mean**	**CI**	**Mean**	**CI**	**Mean**	**CI**
Compulsory	**0.63**	**0.63–0.64**	**0.70**	**0.70–0.70**	**0.67**	**0.67–0.67**
Upper Secondary	**0.64**	**0.64–0.65**	**0.71**	**0.71–0.71**	**0.67**	**0.67–0.68**
University	**0.66**	**0.66–0.67**	**0.72**	**0.71–0.72**	**0.68**	**0.68–0.69**
Foreign	**0.59**	**0.58–0.60**	**0.69**	**0.69–0.70**	**0.66**	**0.65–0.66**
Domestic	**0.65**	**0.65–0.65**	**0.71**	**0.71–0.71**	**0.68**	**0.68–0.68**
**Do you suffer that much impairment from any joint that you are willing to undergo surgery? Yes/No**						
**Education or country of birth**	**Baseline**		**3 months**		**12 months**	
	**OR**	**CI**	**OR**	**CI**	**OR**	**CI**
Compulsory	**1.65**	**1.51–1.81**	**1.36**	**1.21–1.52**	**1.23**	**1.12–1.35**
Upper Secondary	**1.27**	**1.16–1.39**	**1.20**	**1.07–1.35**	**1.12**	**1.03–1.23**
University	1		1		1	
Foreign	**1.34**	**1.19–1.51**	1.09	0.93–1.27	1.07	0.94–1.22
Domestic	1		1		1	
**Does your pain cause you difficulties with walking? Yes/No**						
**Education or country of birth**	**Baseline**		**3 months**		**12 months**	
	**OR**	**CI**	**OR**	**CI**	**OR**	**CI**
Compulsory	1.05	0.97–1.15	**1.12**	**1.03–1.20**	**1.16**	**1.03–1.20**
Upper Secondary	1.05	0.98–1.15	1.02	0.95–1.10	1.06	0.99–1.14
University	1		1		1	
Foreign	**1.13**	**1.0–1.28**	**1.14**	**1.02–1.27**	**1.16**	**1.04–1.30**
Domestic	1		1		1	

Significant results (*p* > 0.05) are indicated in bold. Continuous variables (NRS Pain and EQ-5D-3L) were analyzed with ANCOVA, adjusted mean scores are presented with CI 95%. When analyzing outcomes for different educational groups; educational level was used as a fixed factor, and age, sex, BMI, birthplace were used as covariates in all models. When analyzing outcomes for country of birth; country of birth was used as a fixed factor, and age, sex, BMI, educational level were used as covariates in all models. For the 3 and 12-month follow-up, baseline values were added as covariates in the models. Logistic regression are presented with odds ratios (OR) and confidence intervals (95% CI). Models are controlled for age, sex, BMI in all models, and for baseline values on 3 and 12 months.

**Table 5 jcm-09-02294-t005:** Adjusted outcomes for variables related to health-related behaviours and adherence to the self-management program (*n* = 22,741).

Odds for Being Physical Active more than 150 min Per Week Yes/No						
Education or Country of Birth	Baseline		3 Months		12 Months	
	OR	CI	OR	CI	OR	CI
Compulsory	**0.79**	**0.74–0.85**	**0.79**	**0.74–0.90**	**0.71**	**0.66–0.77**
Upper Secondary	**0.87**	**0.82–0.93**	**0.88**	**0.82–0.95**	**0.85**	**0.79–0.91**
University	1		1		1	
Foreign	**0.74**	**0.66–0.82**	**0.90**	**0.90–1.00**	**0.88**	**0.78–0.98**
Domestic	1		1		1	
**Fear avoidance behaviour: Are you afraid that exercise or physical activity will be harmful to your joints?**						
**Education or country of birth**	**Baseline**		**3 months**		**12 months**	
	**OR**	**CI**	**OR**	**CI**	**OR**	**CI**
Compulsory	1.06	0.96–1.17	**1.40**	**1.19–1.65**	**1.86**	**1.62–2.14**
Upper Secondary	0.97	0.89–1.07	1.15	0.98–1.35	**1.45**	**1.26–1.65**
University	1		1		1	
Foreign	**1.81**	**1.60–2.04**	**2.26**	**1.89–2.69**	**2.50**	**2.16–2.90**
Domestic	1		1		1	
**Odds for number of supervised exercise sessions the patient participated in compared to none** **(0 times reference group)**						
**Education or country of birth**	**1–9 Times**		**10 times or more**			
	**OR**	**CI**	**OR**	**CI**		
Compulsory	**0.82**	**0.76–0.89**	0.97	0.90–1.05		
Upper Secondary	**0.90**	**0.84–0.98**	0.99	0.92–1.07		
University	1		1			
Foreign	**1.26**	**1.12–1.42**	**1.26**	**1.11–1.42**		
Domestic	1		1			
**How often do you use acquired knowledge from the self-management program?** **Every day or several times daily** **(compared to those who answered: Every week, month, never, or don’t know)**						
**Education or ** **country of birth**	**1–9 Times**		**10 times or more**			
	**OR**	**CI**	**OR**	**CI**		
Compulsory	**0.84**	**0.78–0.90**	**0.85**	**0.79–0.91**		
Upper Secondary	0.98	0.92–1.05	**0.91**	**0.85–0.98**		
University	1		1			
Foreign	**1.16**	**1.05–1.29**	**1.20**	**1.08–1.33**		
Domestic	1		1			

Significant results (*p* > 0.05) are indicated in bold. Logistic and Multinominal regression are presented with odds ratios (OR) and confidence intervals (95% CI). Models are controlled for age, sex, BMI in all models, and for baseline values on 3 and 12 months.

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
