# Peer review of "Sociodemographic Inequalities in Outcomes of a Swedish Nationwide Self-Management Program for Osteoarthritis: Results from 22,741 Patients between Years 2008–2017"

_jcm, 2020, doi:10.3390/jcm9072294_

Round 1

Reviewer 1 Report

Please provide a supplementary file in a different format (eg, tar.gz). I couldn’t uncompress the file.

 Please provide a flowchart of sample selection and size.

Please use other methods (eg., Benjamini-Hochberg) for p-value correction for multiple tests which controls false positive better than Bonferroni.

Please provide the models and code for reproducibility.

Please improve manuscript English.

Author Response

Thank you for the review and queries regarding our manuscript. You will see our responses marked in red.

“Please provide a supplementary file in a different format (eg, tar.gz). I couldn’t uncompress the file”.

Authors response: This was initially provided as Figure A1 in the appendix. We have now instead included this in the manuscript instead of the appendix. In addition we have converted the other supplementary files into tar.gz 

“Please use other methods (eg., Benjamini-Hochberg) for p-value correction for multiple tests which controls false positive better than Bonferroni”.

Authors response: We argue that the Bonferroni p-value multiple comparison correction is an accepted method and adequate for our 2 continuous outcomes within the context of our study. Although more conservative than the Benjanmin-Hochberg approach, we found significant results for analyses applying these 2 variables. We don’t have an unreasonably large number of tested hypothesis which would affect type 1 errors to a large extent. Furthermore, considering our large sample, the Bonferroni test does not affect the power of our analyses leaving the Benjamini-Hochberg approach redundant (no difference in finding if Bonferroni or Benjamini-Hochberg approach will result in this case).

“Please provide the models and code for reproducibility”.

Authors response: All data analyses were performed on a Secure Online Data Access (SODA) service provided by the Centre of Registers, Västra Götaland, Sweden where the BOA national registry is housed. The SODA service provides access to confidential national registry data for approved studies, as well as access to statistical programs so that all statistical modeling and outputs are performed and housed within the SODA system and cannot be copied to a remote desktop. In other words, we can only access data as well as the statistical models and codes via our login to the SODA service but this cannot be extracted to external systems. To inform readers, we have now added the following information in the statistics section of the manuscript “All statistical analyses were conducted using SPSS for Windows, version 25. on a Secure Online Data Access (SODA) service provided by the Centre of Registers, Västra Götaland, Sweden where the BOA national registry is housed.”

“Please improve manuscript English”.

Authors response: Our native English author (Allan Abbott) has now made further improvements to the English to aid readability of the manuscript.

Reviewer 2 Report

It is a great work that the authors spent about 10 years and collected as many as 22741 patients to evaluate the relationship between sociodemographic background and different parameters of osteoarthritis. It is really interesting, but I still notice some points which need to be clarified.

Firstly, I notice that the age range of the patients are very broad, from 27 to 95 and the mean age is 66.3. I wonder if the results are same in young people or aged group?

Secondly, I also notice that there are much more female than male. Could the authors explain this and estimate the influence of gender in this studies?

Author Response

Thank you for the review and queries regarding our manuscript. You will see our responses marked in red.

Firstly, I notice that the age range of the patients are very broad, from 27 to 95 and the mean age is 66.3. I wonder if the results are same in young people or aged group?

Authors response: With a mean age of 66.3 years and a standard deviation of ± 9 years, the cases with a min =27 and max=95 are outliers age wise. They are too few to be able to make statistical comparisons with the rest of the group.

Secondly, I also notice that there are much more female than male. Could the authors explain this and estimate the influence of gender in this studies?

Authors response: Gender distribution is in line with national levels for all patients with baseline data for the entire national registry and it exists similar literature reporting the prevalence of symptomatic OA. A recent publication found no association between gender and pain outcome in the BOA registry (Dell’Isola A, Jönsson T, Nero H, Eek F, Dahlberg L. Factors Associated With the Outcome of a First-Line Intervention for Patients With Hip or Knee Osteoarthritis or Both: Data From the BOA Register. Physical Therapy, pzaa113, https://doi.org/10.1093/ptj/pzaa113).

Round 2

Reviewer 1 Report

please review line 52.

Reviewer 2 Report

I have no more comments.